# The role of export competitiveness in driving Renminbi cross-border settlement: Micro-level evidence from China

Danlu Bu[1], Lin Jiang [1,2]*

1 School of Accounting, Southwestern University of Finance and Economics, Chengdu, China, 2 School of Accounting, Sichuan Vocational College of Finance and Economics, Chengdu, China

* jianglin_90@163.com

## Abstract

Cross-border trade settlement in Renminbi (RMB) is a crucial component of China's efforts to internationalize its currency. This study develops micro-level indicators of RMB cross-border settlement to empirically investigate the role of export competitiveness in promoting the use of RMB in international trade. The findings reveal that export competitiveness significantly enhances RMB cross-border settlement, with the establishment of Free Trade Zones further amplifying this effect. Heterogeneity analysis indicates that the impact is more pronounced in firms with smaller scales, higher industry concentration, or greater operating liabilities. Moreover, the increased use of RMB in cross-border settlements, driven by export competitiveness, is associated with reduced import dependence and enhanced firm value. These results deepen the understanding of RMB internationalization from the perspective of firm-level trade activities and provide valuable empirical insights for the strategic promotion of RMB internationalization.

## 1. Introduction

The internationalization of a nation's currency serves as a crucial indicator of its overall economic strength and plays a central role in fostering international exchanges and cooperation. It bolsters a country's global influence and its role in the world monetary system, while also contributes to the stability and security of international trade [1]. In today's globalized economy, currency internationalization has become a dominant trend. As China's official currency, the internationalization of RMB holds the potential to mitigate exchange rate risks and lower transaction costs in cross-border trade, thus facilitating international trade and investment. Moreover, RMB internationalization can spur growth in both China's macroeconomic and microeconomic sectors. In addition to its economic benefits, it plays a key role in safeguarding China's financial security and elevating its national credit and competitiveness on the global stage.

The internationalization of the RMB can be defined as the process by which the currency extends beyond national borders, circulates internationally, and progressively establishes itself as a widely recognized currency for settlement, a unit of account, and a reserve asset in the global economy. RMB settlement refers to the agreement between two parties engaged

**Data availability statement:** All relevant data are within the paper and its Supporting Information files.

**Funding:** The author(s) received no specific funding for this work.

**Competing interests:** The authors have declared that no competing interests exist.

in cross-border trade to use RMB as the settlement currency. For instance, when a Chinese enterprise exports goods to an overseas enterprise, it is agreed that the overseas enterprise will make payment in RMB. RMB valuation refers to the value of international commodities being calculated and denominated in RMB, with the prices of other national currencies being converted according to the exchange rate with RMB. It is also worth noting that the majority of international trade and financial transactions in the international market are now denominated in U.S. dollars. Finally, the term RMB reserves refers to the fact that the RMB is used as the currency of foreign exchange reserves by countries around the world. In the first quarter of 2022, the RMB accounted for 2.88% of global foreign exchange reserves, marking an increase of 1.8 percentage points since 2016, when it was first included in the Special Drawing Rights (SDR) basket of currencies. In terms of functionality, the valuation and reserve currency must be a currency that is widely used internationally for payment and settlement. This underscores the fundamental role of RMB cross-border settlement. Furthermore, international experience has demonstrated that currency internationalization can also entail the local currency acting solely as an international settlement currency. Consequently, RMB internationalization is also directly defined as a certain proportion of international trade transactions settled in RMB [2]. A review of the historical development of RMB internationalization in China reveals that the process of RMB internationalization is a gradual one, progressing from the initial stage of "settlement" to the subsequent stages of "valuation" and then to "reserve". In accordance with the principle of "resolving the easy before the difficult", it is recommended that the objective at this stage be the RMB settlement of cross-border trade [3], thereby promoting the internationalization of the RMB in an orderly manner.

As China's high-level opening to the outside world continues to deepen, the proportion of RMB settlement in cross-border trade has been increasing, making the RMB the most widely used major currency globally [4]. In 2023, the total value of RMB cross-border receipts and payments reached RMB 52.3 trillion, reflecting a year-on-year increase of 24.2%. This has led to the RMB becoming the primary settlement currency for China's cross-border receipts and payments. Specifically, The total amount of cross-border RMB receipts and payments for trade in goods was RMB 10.7 trillion, accounting for 25% of the total cross-border receipts and payments for trade in goods in both local and foreign currencies during the same period. This marks a 7 percentage point increase compared to 2022, indicating notable progress in the adoption of RMB for cross-border trade settlement. Nevertheless, in comparison with internationally dominant currencies such as the US dollar and the euro, the proportion of RMB settlement in cross-border trade remains relatively low. According to data released by the Society for Worldwide Interbank Financial Telecommunication (SWIFT), as of June 2023, the US dollar accounted for 42.02% of the global settlement volume, ranking first, followed by the euro at 31.25%, the British pound at 6.88%, and the Japanese yen at 3.36%. The RMB's share of global settlement volume stands at 2.77%, placing it in fifth position. This remains a significant gap when compared to the US dollar and the euro. Thus, the question of how to increase the ratio of RMB settlement in cross-border trade remains a crucial issue in the process of promoting the internationalization of the RMB in an orderly manner in China.

In studies on the choice of currencies for cross-border settlements, macroeconomic factors are often regarded as the primary drivers of currency internationalization. Factors at the macro level, such as a nation's credibility [5], the stability of monetary policy [6], and political and military power [7], collectively shape a country's overall competitiveness in international trade. These elements play a significant role in promoting the use of the exporting country's currency in cross-border settlements. However, macro-level analyses often neglect firm-level heterogeneity, overlooking how factors like size, financial health, and operational capacity influence settlement decisions. Firms are not passive recipients of macroeconomic forces but

actively shape strategies based on their competitive advantages. For instance, Konka Group, a leading Chinese consumer electronics manufacturer, has faced challenges in recent years, including lagging behind in smart TV and high-end display technologies. This has led to a drop in its domestic market share from 2.25% in 2019 to 0.79% in 2022. Rising raw material prices and labor costs further pressured its profitability, resulting in three consecutive years of negative revenue growth from 2020 to 2022 and a loss in 2022. To address these challenges, Konka increasingly used foreign currencies like the U.S. dollar in cross-border transactions. From 2020 to 2022, the share of its foreign currency-denominated monetary items relative to overseas revenue rose from 5% to 6% and then to 10%, reflecting a strategic adjustment in its settlement practices. In contrast, Zoomlion Heavy Industry enhanced its international competitiveness by advancing product intelligence, digitalization, and establishing a global production and sales network. From 2020 to 2022, the proportion of foreign currency-denominated monetary items relative to its overseas revenue declined annually by 3.93%, 7.31%, and 8.33%, reflecting an increased use of RMB in cross-border trade settlements. This underscores the inadequacy of macro-level factors alone in explaining firms' currency choices, as they overlook the critical role of firm-specific competitiveness. Therefore, understanding the micro-level drivers of firms' decisions to use RMB in cross-border trade, and the role of export competitiveness in this process, constitutes the core focus of this study.

This paper introduces a micro-level index for measuring RMB cross-border settlement and empirically investigates the impact of export competitiveness on RMB usage in cross-border trade. The findings reveal that export competitiveness significantly enhances the adoption of RMB for settlement. These results hold robust across several tests, including alternative measures of explanatory variables, lagging control variables, addressing sample selection bias, and excluding special samples. The mechanism test indicates that the establishment of China's Free Trade Zones (FTZs) can enhance the impact of enterprise export competitiveness on cross-border trade RMB settlement, which reflects the efficacy of the FTZ. Concurrently, the heterogeneity analysis indicates that the impact of enterprises' export competitiveness on cross-border trade settlement in RMB is more pronounced in the sub-sample with smaller scales, higher industry concentration or more operating liabilities. Furthermore, this paper reveals that the increase in the proportion of cross-border trade settlement in RMB due to enterprises' export competitiveness can significantly diminish import dependence and increase the value of enterprises.

The potential contributions of this study are as follows: First, existing research on the relationship between firm-level competitiveness and cross-border trade currency choice has predominantly been qualitative [8–10], with limited quantitative analysis due to constraints related to data availability. This study makes an innovative contribution by quantitatively measuring RMB cross-border settlement at the micro level, providing a clearer picture of firms' cross-border RMB settlement practices. This deepens both academic and practical understanding of RMB cross-border settlement and offers new insights into the trajectory of RMB internationalization in China. Second, the study highlights the role of export competitiveness in driving RMB cross-border settlements. It provides valuable insights for enterprises to optimize resource allocation and improve the efficiency of factor integration, while offering practical guidance for policymakers to promote RMB internationalization in a structured and orderly manner. Third, the paper underscores the importance of FTZs in facilitating RMB cross-border settlement, offering a novel perspective on the interaction between trade policies and currency internationalization, which can inform future policy design.

There are several limitations that warrant further investigation. First, while this paper focuses on cross-border settlement, which is the initial stage of RMB internationalization, it does not delve into the RMB's role in pricing and reserve currency functions. Future research

could expand this analysis to include these advanced stages of RMB internationalization. Second, this study is limited to the Chinese context. A comparative cross-country analysis could provide valuable insights into the varying factors that influence the adoption of RMB for cross-border transactions across different economies, thereby broadening our perspective on global currency dynamics. These future research avenues could deepen our understanding of the mechanisms behind RMB internationalization and the broader implications for global trade and finance.

## 2. Theoretical analysis and hypothesis

In the early stages of cross-border trade development, the primary objective of the two parties involved was the exchange of goods, with the subsequent flow of economic benefits being a significant factor. At this stage, the competition for global monetary sovereignty was not yet a prominent concern, and thus, the convenience of the exporter's currency for cross-border trade settlements became a key consideration for both parties involved in the transaction [11]. This paved the way for the theoretical study of the factors influencing the choice of international trade currencies, which began in the 1970s.

The theory of international trade settlement currency selection focuses on the study of international trade and financial transactions, with an emphasis on the choice of currency as a medium of exchange. In 1973, the Swiss economist Grassman [11] analyzed currency selection data from Sweden's import and export trade with Denmark in 1968 and 1973. His findings revealed that 66% of Sweden's merchandise exports were settled in Swedish currency, while only 25% were settled in Danish currency. Similarly, 59% of Sweden's merchandise imports were settled in Danish currency, while only 26% were settled in Swedish currency. This indicates that export commodities are mainly priced and settled in the exporters' currencies. Grassman's findings led to the formulation of the well-known rule of thumb: the choice of settlement and valuation currencies in international trade is based on the stickiness of commodities to national currencies. This reflects the significant role of the subjective preferences of both trading parties in the selection of settlement currencies during the early stages of cross-border trade.

Nevertheless, as economic globalization deepens and more developing countries engage in global value chains, the explanatory power of the aforementioned rule has diminished. Increased competition among nations has intensified the struggle for monetary sovereignty. Consequently, enterprises in various countries have a preference for settling in their own currencies, whether for exports or imports. This has resulted in the capacity determinism of currency choice in cross-border trade. From a macroeconomic perspective, a country's military strength, economic level, and national credibility collectively underpin its global competitiveness [5–7]. When an exporting country holds a larger market share and stronger export competitiveness, its currency becomes more influential and acceptable in the international market. Therefore, the likelihood of adopting the exporting country's currency for settlement increases [12,13].

A review of the internationalization of major global currencies reveals that a national currency's rise to prominence as a global reserve and settlement currency has been driven by the country's overall strength and competitiveness. Prior to the 18th century, the rise of powers like Portugal, Spain, and the Netherlands saw their minted gold and silver coins become global currencies. From the early 19th century to the mid-20th century, Britain's naval dominance contributed to the pound sterling becoming the leading paper currency, rivaling gold itself. This established the pound as an international currency for over 130 years. Following World War II, the United States, having emerged as the war's largest beneficiary, gradually replaced the UK as the world's leading power. The US dollar then rose to dominance

in the international monetary system. Despite being a major importer, the United States' extensive bilateral trade conducted in dollars further cemented the dollar's global position. Similarly, the Japanese yen and Deutsche Mark became important international currencies due to their respective economies' strength and competitiveness. In the 21st century, China gradually emerged as the world's second-largest economy, with its overall national power continuing to grow. While the RMB still lags behind currencies like the US dollar, its share in global cross-border settlements has steadily increased, demonstrating that the enhancement of national competitiveness has provided strong support for the internationalization of the RMB.

Some might argue that macroeconomic conditions alone, such as China's robust trade surplus, Geopolitical relationships, or global trade policies, are the primary drivers of RMB cross-border settlement, with firms merely adapting to these external circumstances. However, theoretical and empirical evidence challenges this view.

From a theoretical standpoint, firms' economic strength reflect their strategic choices and business decisions in response to similar market conditions. This strength may arise from factors such as firm size [14], market share or product differentiation [15–17], and cost control capabilities [18]. This overall competitive advantage enhances the firm's market dominance in foreign markets, influencing its currency choice and reducing the cost of using the home country's currency for settlement, while also mitigating exchange rate risks [9,10].

In global multilateral economic relations, currencies from developed countries, such as the US dollar and the euro, currently dominate cross-border trade [19]. Even when firms from non-major currency countries engage in cross-border transactions, most tend to settle in third-party currencies [11,20,21], reflecting an inertia in the use of mainstream currencies [22]. Breaking free from this inertia, however, comes with higher transaction costs [23,24]. Currency substitution on a large scale only occurs when the new currency offers a clear advantage [25]. As a result, firms with stronger competitive positions are better equipped to overcome the inertia of using dominant international currencies [26]. First, competitive firms often offer superior products or services, which incentivizes importers to agree to settlement in the exporter's currency. For instance, Buyers may prioritize transactions in the exporter's currency in order to obtain higher-quality goods, rather than insisting on using traditional trade currencies. This dynamic aligns with the broader notion of "pricing-to-market", where competitive exporters can influence not only prices but also payment terms, including the choice of settlement currency [27]. Second, firms with strong competitiveness typically have diversified global supply chains and sales networks, which reduce the correlation of revenues across different branches, and lower the risk of expected returns [28], thus reducing reliance on foreign currencies.

In practice, as China's economic strength continues to grow, more and more Chinese firms are using the RMB for cross-border trade settlements. In May 2023, Midea Payment Technology Co., Ltd. became the first third-party payment institution in Shenzhen to obtain cross-border RMB qualifications, facilitating the broader development of RMB settlements in Midea Group's international trade. Similarly, companies like ZTO Express have expanded their RMB cross-border settlement services based on their own strong foundations. The footprint of Chinese corporations in international markets has been more and more significant, with the size of these transactions prompting many to reassess the likely pace of the usage of RMB as an alternate vehicle currency [29].This indicates that, supported by national competitiveness at the macro level, the export competitiveness of enterprises, driven by their resource and capability advantages at the microeconomic level, forms the internal driving force for gaining control over currency choice in international trade.

Based on the preceding analysis, this paper posits the following hypothesis:

H1: Enterprise export competitiveness has a positive effect on promoting RMB cross-border settlement.

## 3. Methodology

The construction of RMB cross-border settlement indicators in this study requires the use of foreign currency monetary items. However, the limitations caused by undisclosed, incomplete, or inconsistently reported foreign currency monetary items in the annual reports of listed companies prior to 2013 constrained data collection. In 2014, the China Securities Regulatory Commission (CSRC) revised the *Information Disclosure and Reporting Rules for Companies with Publicly Issued Securities No. 15 - General Provisions on Financial Reporting*, This revision standardized information disclosure requirements, enhanced integration, and improved the comprehensiveness of foreign currency monetary item disclosures in annual reports, thereby increasing the accessibility of foreign currency data.

As a result, this study selects all A-share listed companies from 2014 to 2022 as the initial sample, and applies the following processing steps: (1) excluding companies without overseas business revenue, as the measurement of RMB cross-border settlement requires the use of overseas business revenue as the denominator; (2) excluding the sample of listed companies in the financial industry, as their unique business models and regulatory frameworks differ significantly from those of non-financial firms; and (3) excluding the sample of listed companies with incomplete data. The final sample consists of 2,081 listed companies with a total of 10,026 observations for nine years.

Foreign currency monetary item data, essential for calculating RMB cross-border settlement, are manually extracted from company annual reports. Data for other variables are obtained from the CSMAR and CNRDS databases. To ensure the stability of the model, all continuous variables are winsorized at the upper and lower 1% levels.

The dependent variable in this paper is RMB cross-border settlement. To measure this, we focus on the enterprise's overseas sales and collection cycle. When payment is received, the overseas business revenue is deposited into the money fund account, and when payment is not received, it is recorded as accounts receivable. Therefore, we define the foreign currency settlement ratio (*FC*) as the sum of the increase in the amount of foreign currency in both the money fund and accounts receivable, divided by the current year's overseas business revenue. This approach provides a comprehensive reflection of the proportion of foreign currency settlements within the overseas business revenue. And it also ensures that the changes in foreign currency settlement are directly linked to the revenue generated in the same period, allowing for a better assessment of the settlement behavior over time. This indicator is a negative measure, meaning a higher FC value indicates a lower degree of RMB cross-border settlement.

The independent variable in this paper is export competitiveness (*COMP*). Liu et al. [30] proposed that a firm's resources and capabilities are two key aspects of measuring its international competitiveness. Since then, scholars have primarily focused on financial indicators to evaluate the export competitiveness of firms from these two dimensions. Based on the previous literature, this study selects the following indicators and applies the entropy weighting method to combine them, thereby reflecting the export competitiveness of firms: **(1) Total assets**. Assets, as the most direct representation of firm resources, intuitively reflect the resource-based competitive advantage of firms in export activities; **(2) Proportion of R&D personnel**. Modern competition among enterprises ultimately revolves around talent, with human resources being a critical driver of competitiveness. General Secretary Xi emphasized that "the key to doing a good job in China is in the Party, the key is in the people, and the key is in the talents". Thus, this paper uses the ratio of R&D personnel to total personnel as an indicator of the resource-based export competitiveness of firms; **(3) Asset-liability ratio**. Financing constraints are a significant barrier to enterprise development. The ability of firms to access sufficient financing for international operations and expansion is a key determinant of their international competitiveness. This study uses the asset-liability ratio as a measure

of financing capacity, reflecting the capability-based export competitiveness of firms; **(4) Profitability**. Strong profitability is essential for a firm's survival and a key factor in gaining a competitive edge in international market. Therefore, this paper considers profit margin as an indicator of the capability-based export competitiveness of firms; **(5) Growth rate of operating income**. Revenue growth is a clear indicator of a firm's continuous development and expansion. This study uses the growth rate of operating income to measure the firm's developmental capacity, which contributes to assessing its export competitiveness; **(6) Market share**. The proportion of market share a firm holds is a crucial indicator of its competitiveness, reflecting its market influence and industry positioning. This paper adopts the ratio of a firm's operating income to the total operating income of all firms in the same industry as a measure of market share, representing the capability-based export competitiveness of firms; **(7) Cost advantage**. A firm's cost advantage is intrinsically linked to its export competitiveness. Effective cost management plays a pivotal role in enterprise development, and firms with advanced cost management capabilities can gain a competitive edge in increasingly complex international trade [18]. Based on Poter [18], this paper uses the ratio of operating income to operating costs as a proxy for cost advantage, reflecting the capability-based export competitiveness of firms.

In this paper, a range of control variables are selected to account for various firm-specific characteristics and macroeconomic factors. These include firm age (*AGE*), accounts receivable turnover (*ART*), capital intensity (*KL*), return on assets (*ROA*), tradable shares ratio (*TS*), cash flow ratio (*CF*), type of ownership (*SOE*), exchange rate (*ER*), government subsidy ratio (*SUB*), and the proportion of overseas business income (*OI*). The specific definitions of each variable are detailed in S1 Appendix. Specifically, firm age (*AGE*) captures the potential effect of maturity on business stability and international operations, while accounts receivable turnover (*ART*) and capital intensity (*KL*) reflect a firm's operational efficiency and asset structure, both of which may impact cross-border trade and currency choice. Return on assets (*ROA*) and cash flow ratio (*CF*) are included to account for financial health, which could influence a firm's ability to handle foreign exchange risks and settle trade in RMB. The tradable shares ratio (*TS*) and ownership type (*SOE*) are included to control for the influence of ownership structure on a firm's international behavior.

To account for omitted variables at the firm and macroeconomic levels, a panel data model is employed as in equation (1), with both individual (*ID*) and year (*YEAR*) fixed effects. This allows us to control for unobserved heterogeneity and potential year-specific shocks. Considering that the impact of firms' export competitiveness on RMB cross-border settlement may have a time lag, and also to avoid the endogenous effect of reverse causality, this paper substitutes export competitiveness lagged by one period as an independent variable in the model. The coefficient of export competitiveness (*COMP*) is expected to be significantly negative if the hypothesis holds, indicating that stronger export competitiveness is associated with a higher degree of RMB cross-border settlement.

$$FC_{it} = \alpha + \beta \cdot COMP_{i,t-1} + \sum controls_{it} \cdot \gamma + YEAR_t + ID_i + \varepsilon_{it} \qquad (1)$$

## 4. Empirical results

### 4.1 Summary statistics

Table 1 shows the descriptive statistics of the variables. The results show that, in the full sample, the mean value of *FC* is 0.025, with a minimum of −5.691 and a maximum of 3.968. This suggests that the proportion of foreign currency settlements among listed companies is

relatively evenly distributed, with the average proportion of foreign currency settlements at the firm-year level being 2.5%. In addition, the mean value of export competitiveness (*COMP*) is 0.21, with a minimum of 0.167 and a maximum of 0.303, indicating some variation in export competitiveness across firms. Descriptive statistics for the remaining control variables are also provided.

Fig 1 shows the annual trend in the proportion of foreign currency settlements of enterprises. The data reveal that from 2014 to 2022, the proportion of foreign currency settlements exhibited a gradual decline. In 2014, RMB cross-border settlement was in its early stages of initiation and rapid development, with approximately 25% of China's cross-border trade settled in RMB. As a result, the proportion of foreign currency settlements was relatively low in 2014. On August 11, 2015, China implemented an RMB exchange rate reform, abandoning the policy of pegging the RMB to the US dollar in favor of an exchange rate system referencing a basket of currencies. Influenced by the market volatility caused by the reform, the subsequent depreciation of the RMB and the slowdown in capital account opening, as well as

**Table 1. Summary statistics.**

| Variables | Observations | Mean | Std | Min | Max |
|---|---|---|---|---|---|
| *FC* | 10026 | 0.025 | 0.903 | −5.691 | 3.968 |
| *COMP* | 10026 | 0.210 | 0.025 | 0.167 | 0.303 |
| *AGE* | 10026 | 2.951 | 0.283 | 2.197 | 3.638 |
| *KL* | 10026 | 2.227 | 1.420 | 0.451 | 9.231 |
| *TS* | 10026 | 0.771 | 0.218 | 0.247 | 1 |
| *ROA* | 10026 | 0.029 | 0.085 | −0.414 | 0.204 |
| *ART* | 10026 | 2.488 | 1.709 | 0.905 | 12.258 |
| *CF* | 10026 | 0.050 | 0.064 | −0.132 | 0.236 |
| *SOE* | 10026 | 0.209 | 0.407 | 0 | 1 |
| *ER* | 10026 | 6.719 | 0.248 | 6.124 | 7.013 |
| *SUB* | 10026 | 0.014 | 0.016 | 0.000 | 0.095 |
| *OI* | 10026 | 0.234 | 0.238 | 0.000 | 0.939 |

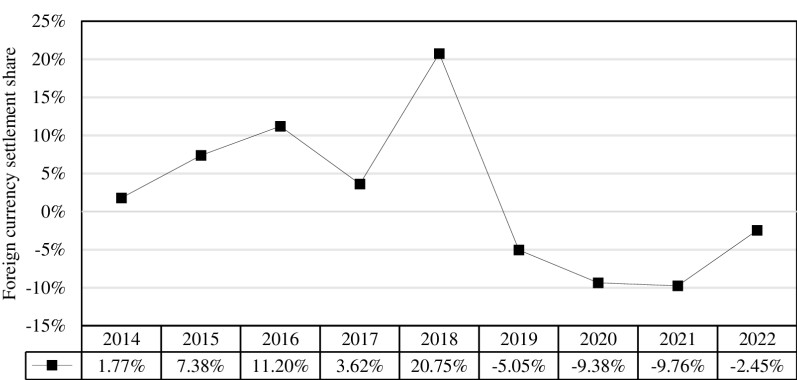

| | 2014 | 2015 | 2016 | 2017 | 2018 | 2019 | 2020 | 2021 | 2022 |
|---|---|---|---|---|---|---|---|---|---|
| ■ | 1.77% | 7.38% | 11.20% | 3.62% | 20.75% | -5.05% | -9.38% | -9.76% | -2.45% |

The data for each year in the graph are calculated as the arithmetic mean of the data for all sample companies in that year.

**Fig 1. The annual trend in average foreign currency settlement share.**

the trade friction between China and the US in 2018, the development of RMB cross-border settlement in China was in a relatively stagnant stage from 2015 to 2018, resulting in a lower proportion of foreign currency settlements during this period. following 2019, China accelerated its financial opening, strongly promoted the international use of RMB through initiatives such as the signing and updating of bilateral currency swap agreements, and collaborated with the Bank for International Settlements to establish the RMB liquidity arrangement, among other measures. These efforts contributed to an expansion in the proportion of RMB settlements in cross-border trade, marking the beginning of a period of rapid development for RMB cross-border settlement. Fig 1 shows that in 2019, the proportion of foreign currency settlement was −5.05%, representing the first instance of negative growth in foreign currency settlement. From 2020 to 2022, the proportion of foreign currency settlements remained at a relatively low level, suggesting that the momentum for RMB cross-border settlement among Chinese enterprises remains favourable.

The annual trend of RMB cross-border settlement by enterprises, as reflected in this micro-level data, aligns with the macro-stage progression of RMB internationalization in China. This pattern underscores that RMB cross-border settlement is not only the foundation but also the initial step in advancing RMB internationalization. Moreover, it validates the feasibility and validity of the measurement framework for RMB cross-border settlement constructed in this paper, thereby providing a solid data foundation for the research.

## 4.2 Regression analysis

The basic regression results are presented in Table 2. Columns (1)-(3) display the results of the uni-variate regression, the regression controlling for only year and individual fixed effects, and the regression with all control variables included, respectively. These results indicate that export competitiveness is significantly and negatively correlated with the share of foreign currency settlement, with all coefficients being statistically significant at the 5% level. The coefficient of *COMP* in column (3) shows that for each unit increase in a firm's export competitiveness, the share of foreign currency settlement decreases by 226.5%. This suggests that higher export competitiveness significantly promotes RMB cross-border settlement, thus supporting the hypothesis of this paper.

## 4.3 Robustness test

**4.3.1 Replacement of explanatory variable measures.** Based on the overseas sales and collection cycle, this paper proposes a measure of RMB cross-border settlement, calculated as the ratio of the increase in foreign currency accounts receivable and foreign currency monetary funds to overseas business revenue, which is reasonable and feasible. However, considering that part of the increase in foreign currency monetary funds may originate from the company's separate foreign currency reserves rather than overseas business income, this paper further measures RMB cross-border settlement by using the ratio of the increase in foreign currency accounts receivable alone to overseas business income (*FC_new*) and re-runs the regression.

The results presented in column (1) of Table 3 show that the coefficient of *COMP* remains significantly negative, confirming that export competitiveness indeed promotes RMB cross-border settlement.

**4.3.2 Control variables lagged by one period.** To mitigate the potential issue of reverse causality between the control variables and the dependent variable, this paper lags all other control variables by one period and re-runs the regression.

The regression results are shown in column (2) of Table 3, where the coefficient on *COMP* remains negative and significant at the 10% level. It suggests that, even after accounting for the

**Table 2. The impact of export competitiveness on RMB cross-border settlement.**

|  | FC | | |
| --- | --- | --- | --- |
|  | (1) | (2) | (3) |
| COMP | −1.144** | −2.130** | −2.265** |
|  | (−2.50) | (−2.26) | (−2.40) |
| AGE |  |  | −0.224 |
|  |  |  | (−0.84) |
| KL |  |  | −0.002 |
|  |  |  | (−0.09) |
| TS |  |  | −0.127 |
|  |  |  | (−1.57) |
| ROA |  |  | 0.942*** |
|  |  |  | (4.72) |
| ART |  |  | −0.033 |
|  |  |  | (−1.47) |
| CF |  |  | 0.102 |
|  |  |  | (0.45) |
| SOE |  |  | −0.327*** |
|  |  |  | (−3.65) |
| ER |  |  | 0.093 |
|  |  |  | (0.46) |
| SUB |  |  | −2.619* |
|  |  |  | (−1.83) |
| OI |  |  | 0.467*** |
|  |  |  | (3.34) |
| Year FE | no | yes | yes |
| Firm FE | no | yes | yes |
| N | 10026 | 10026 | 10026 |
| R² | 0.002 | 0.015 | 0.028 |
| F | 6.26** | 13.20*** | 8.50*** |

Robust standard error adjusted t-values are shown in brackets. *, **, *** denote 10%, 5% and 1% levels of significance respectively.

time lag of all variables, the export competitiveness of firms continues to significantly promote RMB cross-border settlement, indicating that the conclusion of this paper is robust.

**4.3.3 Adjustment for the effect of sample selection.** The measurement of RMB cross-border settlement indicators requires the use of overseas business revenue, meaning that the sample in this paper consists solely of enterprises with overseas business revenue, which may introduce selectivity bias into the results. To address this potential bias, the two-stage Heckman procedure is employed.

Column (3) in Table 3 presents the result of the first stage, where a probit model is used to estimate the probability that enterprises generate overseas business income (*SELECT*), incorporating all control variables in this paper as well as the amount of enterprises' overseas investment (*INV*). Column (4) shows the result of the second stage, which is obtained by calculating the Inverse Mills Ratio (*IMR*) based on the first-stage regression estimates and adding it to the original model as a control variable. The results indicate that, after correcting for selectivity bias, the export competitiveness of enterprises continues to have a significant positive effect on RMB cross-border settlement.

**4.3.4 Special sample exclusion.** In 2015, China implemented a significant RMB exchange rate reform, reversing the policy of pegging the RMB to the US dollar and allowing the

**Table 3. Robustness test.**

| | FC_new | FC | SELECT | FC | FC |
|---|---|---|---|---|---|
| | (1) | (2) | (3) | (4) | (5) |
| COMP | −0.650** (−2.14) | −1.692* (−1.75) | | −2.318** (−2.45) | −2.314** (−1.98) |
| AGE | −0.116 (−1.31) | −0.200 (−0.82) | 0.123*** (4.45) | 0.033 (0.28) | −0.294 (−0.91) |
| KL | −0.011 (−1.64) | −0.019 (−0.93) | −0.089*** (−19.97) | 0.014 (0.48) | −0.003 (−0.11) |
| TS | −0.057** (−2.46) | −0.131* (−1.77) | 0.651*** (20.13) | −0.212** (−1.96) | −0.122 (−1.31) |
| ROA | 0.283*** (4.24) | 0.386** (2.16) | −0.254** (−2.36) | 1.007*** (4.83) | 1.060*** (4.63) |
| ART | −0.024*** (−3.60) | −0.021 (−1.02) | −0.055*** (−14.06) | −0.005 (−0.41) | −0.027 (−0.95) |
| CF | −0.239*** (−3.10) | 0.271 (1.27) | 0.277** (2.27) | 0.033 (0.15) | −0.011 (−0.04) |
| SOE | −0.051** (−2.01) | −0.344*** (−3.82) | −0.357*** (−18.73) | −0.288*** (−2.96) | −0.333*** (−3.47) |
| ER | 0.052 (0.48) | 0.223 (0.33) | 2.046*** (20.45) | −0.301 (−0.96) | 0.122 (0.53) |
| SUB | −0.821** (−2.16) | −0.499 (−0.48) | 3.125*** (7.22) | −2.794* (−1.94) | −2.291 (−1.47) |
| OI | 0.220*** (4.89) | −0.008 (−0.08) | 1.490*** (39.43) | 0.271 (1.50) | 0.467** (2.51) |
| INV | | | −0.016*** (−19.23) | | |
| IMR | | | | −0.167 (−1.02) | |
| Year FE | yes | yes | yes | yes | yes |
| Firm FE | yes | yes | yes | yes | yes |
| N | 10026 | 10026 | 32767 | 10026 | 8683 |
| R² | 0.037 | 0.021 | 0.153 | 0.027 | 0.028 |
| F(Wald-chi²) | 10.78*** | 8.29*** | 3605.69*** | 7.98*** | 7.62*** |

Robust standard error adjusted t-values are shown in brackets. *, **, *** denote 10%, 5% and 1% levels of significance respectively. Each control variable in column (2) is lagged by one period. Column (3) includes 32,767 observations due to the incorporation of additional unselected data using a probit model, so the R² is reported as a pseudo R², and the last row presents the Wald chi-squared statistic. In column (5), some samples are excluded, reducing the observations to 8,683.

exchange rate to float within a specified range. In the same year, the RMB was included in the International Monetary Fund's Special Drawing Rights (SDR) currency basket, marking its emergence as an international reserve currency. As a result, the volume of RMB cross-border settlement saw a substantial increase in 2015. Additionally, in 2018, the trade friction between the United States and China escalated, with the United States imposing tariffs on certain Chinese goods, which had a significant impact on specific industries. This created an abnormal trend in RMB cross-border settlement.

Therefore, to mitigate the impact of the special event in 2015 and specific industries in 2018, this paper re-estimates the regressions by excluding the entire 2015 sample and specific industry samples affected by the 2018 tariffs. The results in column (5) of Table 3 confirm that the conclusions of this paper remain robust.

## 4.4 Mechanism

To overcome the disadvantages faced by multinational enterprises in international business, the support of the home country government is of great importance, which can help enterprises transform disadvantages into advantages, thereby enhancing their international competitiveness [31]. Free trade zones (FTZs) serve as a key lever for China in developing a high degree of openness to the outside world. They play a pioneering role in promoting the liberalization and facilitation of foreign trade and are a significant initiative by the Chinese government to support export-oriented enterprises. By 2022, China has established FTZs in 21 provinces and municipalities, including Shanghai and Guangdong, offering enterprises greater advantages and convenience in conducting export business.

Therefore, this paper identifies whether an enterprise is located in an FTZ and constructs a cross-multiplier of the dummy variable for FTZ establishment (*ZONE*) and export competitiveness (*COMP\*ZONE*). This cross-multiplier term is then added to the original model to re-run the regression. The results shown in column (1) of Table 4 reveal that the coefficient of the cross-multiplier term is significantly negative, indicating that the establishment of FTZs strengthens the positive effect of export competitiveness on RMB cross-border settlement.

## 4.5 Heterogeneity analysis

**4.5.1 Heterogeneity of firm size.** To further investigate the heterogeneity of the impact of export competitiveness on RMB cross-border settlement, this paper conducts a subsample analysis based on firm size to reveal whether smaller or larger firms are more responsive to the advantages of RMB settlement when improving their export competitiveness.

To test this, firms are divided into two groups based on their total assets, using the median as the cutoff. Firms with total assets above the median are classified as large firms, while those below the median are considered small firms. Subsample regressions are then conducted for each group. The results are shown in columns (2)-(3) of Table 4. It can be seen that the effect of export competitiveness on RMB cross-border settlement is more significant in the small-firm group than in the large-firm group. It suggests that firm size plays a crucial role in moderating the relationship between export competitiveness and RMB cross-border settlement. Smaller firms, likely constrained by limited resources and bargaining power, are more sensitive to the benefits of RMB settlement when they enhance their export competitiveness. This highlights the importance of firm-level microeconomic factors in understanding currency choice in cross-border trade, particularly for smaller market participants.

**4.5.2 Heterogeneity of industry concentration.** The degree of industry concentration reflects the extent to which leading firms in the industry control the market, serving as a measure of the market structure of the industry. It indicates the level of competition and monopoly within the market. A high degree of industry concentration suggests the presence of a few large enterprises that dominate the industry's development. These firms may have a pioneering role in driving export competitiveness and RMB cross-border settlement. Accordingly, this paper employs the industry Lerner index to measure industry concentration and divides the sample into two groups -high and low concentration- based on whether the index is above or below the median. Group regressions are then conducted.

The regression results are shown in columns (4)-(5) of Table 4. The coefficient for *COMP* is significantly negative in the high-concentration group, while it is not significant in the low-concentration group. The coefficient difference test between the two groups reveals a significant disparity, indicating that export competitiveness has a stronger effect on promoting RMB cross-border settlement in industries with higher concentration. This finding highlights

**Table 4. Mechanistic and heterogeneity analyses.**

| | FC | FC | FC | FC | FC | FC | FC |
|---|---|---|---|---|---|---|---|
| | | Large-firm | Small-firm | High-concentration | Low-concentration | Financial liability | Operating liability |
| | (1) | (2) | (3) | (4) | (5) | (6) | (7) |
| COMP | −0.793 (−0.67) | −0.384 (−0.26) | −5.205*** (−3.39) | −4.667** (−2.29) | −0.393 (−0.29) | 0.379 (0.26) | −3.985*** (−2.88) |
| ZONE | 0.520*** (2.68) | | | | | | |
| COMP*ZONE | −2.290** (−2.40) | | | | | | |
| AGE | −0.234 (−0.88) | 0.147 (0.36) | −0.545 (−1.16) | −0.865 (−1.39) | −0.715 (−1.50) | −0.442 (−1.12) | 0.072 (0.18) |
| KL | −0.001 (−0.06) | 0.002 (0.07) | −0.011 (−0.24) | −0.059 (−1.37) | 0.050 (1.44) | −0.024 (−0.79) | −0.019 (−0.45) |
| TS | −0.121 (−1.49) | −0.002 (−0.02) | −0.206 (−1.47) | −0.200 (−1.34) | −0.061 (−0.45) | −0.098 (−0.85) | −0.115 (−0.87) |
| ROA | 0.934*** (4.69) | 1.021*** (3.98) | 0.939*** (3.39) | 0.971*** (2.98) | 0.862*** (3.72) | 0.830*** (3.12) | 1.121*** (3.40) |
| ART | −0.032 (−1.45) | −0.023 (−1.35) | −0.085 (−1.27) | −0.045 (−0.82) | −0.003 (−0.10) | −0.075** (−2.56) | −0.005 (−0.13) |
| CF | 0.106 (0.47) | 0.782** (2.18) | −0.497 (−1.61) | 0.203 (0.60) | −0.288 (−0.81) | 0.399 (1.20) | −0.054 (−0.17) |
| SOE | −0.329*** (−3.67) | −0.346** (−2.57) | −0.350** (−2.56) | −0.411** (−2.19) | −0.252 (−1.63) | −0.229** (−2.55) | −0.342* (−1.85) |
| ER | 0.026 (0.13) | −0.340 (−1.55) | 0.691 (1.55) | 0.864 (0.91) | 0.356 (1.63) | 0.138 (0.69) | 0.002 (0.01) |
| SUB | −2.697* (−1.89) | −3.545 (−1.62) | −1.577 (−0.72) | −5.835** (−2.35) | 1.796 (1.02) | −3.218* (−1.86) | −0.895 (−0.40) |
| OI | 0.475*** (3.42) | 0.240 (1.21) | 0.676*** (2.84) | 0.222 (1.17) | 0.720** (2.45) | 0.501*** (2.62) | 0.063 (0.44) |
| Year FE | yes | yes | yes | yes | yes | yes | yes |
| Firm FE | yes | yes | yes | yes | yes | yes | yes |
| N | 10026 | 5013 | 5013 | 5013 | 5013 | 5013 | 5013 |
| R² | 0.029 | 0.025 | 0.041 | 0.032 | 0.029 | 0.028 | 0.033 |
| F | 8.35*** | 4.40*** | 5.06*** | 3.27*** | 3.53*** | 3.96*** | 4.56*** |
| Diff | | 4.821*** | | −4.274* | | 4.364* | |
| p-value | | (0.000) | | (0.100) | | (0.100) | |

P-values are shown in the brackets of differences in coefficients between groups, and robust standard error adjusted t-values are shown in other brackets. *, **, *** denote 10%, 5% and 1% levels of significance respectively.

the spillover effect of industry leaders in driving the expansion of RMB cross-border settlement across the entire industry.

**4.5.3 Heterogeneity of liability structure.** The liabilities of an enterprise can be classified according to their purpose and source into financial liabilities, which refer to borrowings from financial institutions, the issuance of bonds, or other forms of debt that typically require the payment of high interest rates, and operating liabilities, which refer to obligations arising from the day-to-day operations of the enterprise, such as trade credit from the purchase of raw materials or the acquisition of machinery and equipment. A company's liability structure reflects its preference for either endogenous or exogenous financing. If a company's financial liabilities substantially exceed its operating liabilities, this may indicate an over-reliance on

borrowed capital, thereby placing greater pressure on its operating cash flow. Conversely, higher operating liabilities, which are primarily due to commercial credit, suggest a stronger market position within the industry [32]. Therefore, this paper measures the liability structure of firms by the ratio of financial liabilities to total liabilities, and the group regression is conducted by dividing the sample into two groups: financial liability type and operating liability type, according to whether the ratio of financial liabilities exceeds the median.

The results are reported in columns (6)-(7) of Table 4. The coefficient of *COMP* is significantly negative in the operating liability type enterprises but insignificant in the financial liability type enterprises, and the difference in coefficients between the two groups is statistically significant. The results indicate that the promoting effect of export competitiveness on RMB cross-border settlement is more pronounced in firms with higher operating liabilities, emphasizing the importance of reducing reliance on financial liabilities and strengthening the accumulation of operating cash flow.

## 5. Further analyses

### 5.1 Further analysis of import dependence

In the current global production network, the division of labour among suppliers at all levels is highly specialized, which brings efficiency to the deep integration of global trade but also increases the degree of import dependence [33]. For example, China's manufacturing sector exhibits a high degree of import dependence, particularly in high-end parts and components, which presents an obvious competitive disadvantage. This issue is even more pronounced in agriculture, where high import dependence increases the risk of food shortages due to external shocks [34].

RMB cross-border settlement reflects the competitiveness of enterprises in the international market, and this competitiveness should enhance the flexibility and options available in foreign trade, while also mitigating the supply chain risks associated with import dependence. Given the challenges in collecting import data, this paper measures the import dependence of enterprises using overseas business cost (*DEPEND*) and treats it as the dependent variable, and conducts a linear regression with the cross-multiplier term of export competitiveness and the inverse of foreign currency settlement share (*COMP\*FC_r*) as the independent variable.

The regression results presented in column (1) of Table 5 show that the coefficient of the cross-multiplier term is significantly negative, indicating that after the promotion of RMB cross-border settlement driven by export competitiveness, enterprises reduce their import dependence. This, in turn, enhances their ability to cope with risks and bolsters their supply chain resilience.

### 5.2 Further analysis of firm value

In the capital market, investors' confidence and investment decisions regarding a listed company are ultimately reflected in the company's value through changes in its stock price. Strong export competitiveness and the resulting increase in the level of RMB cross-border settlement means that the company is also competitive in the capital market and may generate higher future earnings. As a result, investors are likely to have an optimistic outlook on the company's stock, driving up its firm value.

In this paper, the firm's Tobin's Q value (*TobinQ*) is used as a proxy for firm value, with the cross-multiplier term (*COMP\*FC_r*) serving as the independent variable to conduct a linear regression to further explore the impact of export competitiveness and RMB cross-border settlement on firm value. The results in column (2) of Table 5 show that the coefficient of

**Table 5. Further analyses.**

|  | DEPEND | TobinQ |
|---|---|---|
|  | (1) | (2) |
| COMP*FC_r | −0.648** | 0.074* |
|  | (−2.57) | (1.81) |
| AGE | 5.027** | −0.092 |
|  | (2.25) | (−0.18) |
| KL | −0.185** | −0.117*** |
|  | (−2.10) | (−6.01) |
| TS | −0.132 | 1.537*** |
|  | (−0.24) | (15.38) |
| ROA | −1.085 | 0.919*** |
|  | (−1.34) | (5.57) |
| ART | −0.024 | 0.003 |
|  | (−0.23) | (0.09) |
| CF | −0.291 | 1.062*** |
|  | (−0.26) | (5.45) |
| SOE | −0.008 | −0.112* |
|  | (−0.02) | (−1.86) |
| ER | 5.684** | −0.752** |
|  | (2.11) | (−2.17) |
| SUB | −1.159 | 1.747 |
|  | (−0.20) | (1.35) |
| OI | 8.359*** | −0.094 |
|  | (9.87) | (−0.57) |
| Year FE | yes | yes |
| Firm FE | yes | yes |
| N | 10026 | 9873 |
| R² | 0.043 | 0.332 |
| F | 9.76*** | 102.05*** |

Robust standard error adjusted t-values are shown in brackets. *, **, *** denote 10%, 5% and 1% levels of significance respectively. There are 153 missing observations for TobinQ in column (2), resulting in a reduced sample size of 9,873 observations.

the cross-multiplier term is significantly positive, indicating that the increase in RMB cross-border settlement driven by export competitiveness can significantly enhance firm value. This facilitates greater access to capital market support, promotes the development of the real economy, and creates a virtuous circle.

## 6. Conclusions

Based on the micro measurement of the degree of RMB cross-border settlement, this paper examines the impact of enterprises' export competitiveness on RMB cross-border settlement. The results show that stronger export competitiveness significantly promotes the use of RMB in cross-border trade, highlighting the critical role of firm-level strength as a micro-level factor in gaining currency autonomy in bilateral trade. Robustness tests, including changes in the measurement of explanatory variables, lagging control variables, and addressing sample selection biases, further confirm the generalizability and reliability of this effect.

Mechanism analysis reveals that the development of FTZs, through improved business environments and policy support, significantly enhances the effect of export competitiveness on RMB settlement in cross-border trade. This underscores the important role of national strategic policies in advancing RMB internationalization, while also demonstrating the potential for synergy between enterprises and policy initiatives.

Heterogeneity analysis shows that the impact of export competitiveness on RMB settlement is more pronounced in certain contexts. For instance, smaller firms, with their flexible resource allocation and quick market responsiveness, tend to experience a stronger effect of competitiveness on RMB settlement. In highly concentrated industries, large leading firms play a crucial role in driving RMB settlement, amplifying the effect of competitiveness. Additionally, firms with higher operating liabilities, by reducing reliance on financial leverage, enhance their flexibility, further strengthening the role of competitiveness in promoting RMB settlement in cross-border trade.

Moreover, the increased use of RMB for cross-border settlements brings several positive spillover effects: (1) It significantly reduces firms' dependence on imports, thereby improving supply chain resilience and risk management, contributing to the long-term stability of their competitiveness. (2) It enhances firm value, fostering a virtuous cycle between the capital markets and the real economy. This underscores the strategic significance of RMB settlement in enhancing firms' global competitiveness and investment appeal.

Based on the conclusions drawn from this study, the following suggestions are proposed. First, enterprises should strengthen their export competitiveness. Enterprises especially small or medium size enterprises (SME) should focus on improving product quality, service differentiation, and operational efficiency. By enhancing these aspects, firms can not only gain a competitive edge in international markets but also facilitate the use of RMB as a preferred settlement currency. Second, policy support for FTZs should be enhanced. The study demonstrates the significant role that FTZs play in strengthening the relationship between export competitiveness and RMB settlement. Policymakers should continue to support FTZ development by improving the business environment and providing targeted incentives for enterprises. This would further encourage the use of RMB in cross-border transactions, particularly among firms operating in FTZs. Third, tailored policies for firms of different sizes and industries should be developed. As the effect of export competitiveness on RMB settlement varies across firms with different characteristics, policymakers should design targeted strategies that consider factors such as firm size, industry concentration, and financial structure. Smaller firms, for instance, may require more flexible policies, while large, industry-leading firms can be leveraged as role models to drive broader RMB adoption within their sectors.

## Supporting information

**S1 Appendix. Variables definition.**
(DOCX)

**S1 File. Data.**
(XLSX)

## Author contributions

**Conceptualization:** Danlu Bu.

**Data curation:** Lin Jiang.

**Investigation:** Lin Jiang.

**Methodology:** Danlu Bu, Lin Jiang.

**Project administration:** Danlu Bu.

**Software:** Lin Jiang.

**Supervision:** Danlu Bu.

**Writing – original draft:** Lin Jiang.

**Writing – review & editing:** Danlu Bu, Lin Jiang.

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
