## [Decision Letter · Decision Letter 0]

29 Nov 2024

PONE-D-24-43217The role of export competitiveness in driving Renminbi cross-border settlement: Micro-level evidence from ChinaPLOS ONE

Dear Dr. JIANG,

Thank you for submitting your manuscript to PLOS ONE. After careful consideration, we feel that it has merit but does not fully meet PLOS ONE’s publication criteria as it currently stands. Therefore, we invite you to submit a revised version of the manuscript that addresses the points raised during the review process.

We look forward to receiving your revised manuscript.

Kind regards,

Xiaoyong Zhou, Ph.D.

Academic Editor

PLOS ONE

Reviewers' comments:

Reviewer's Responses to Questions

**Comments to the Author**

1. Is the manuscript technically sound, and do the data support the conclusions?

Reviewer #1: Partly

Reviewer #2: Partly

Reviewer #3: Yes

2. Has the statistical analysis been performed appropriately and rigorously? 

Reviewer #1: No

Reviewer #2: N/A

Reviewer #3: Yes

3. Have the authors made all data underlying the findings in their manuscript fully available?

Reviewer #1: Yes

Reviewer #2: Yes

Reviewer #3: Yes

4. Is the manuscript presented in an intelligible fashion and written in standard English?

Reviewer #1: No

Reviewer #2: Yes

Reviewer #3: Yes

5. Review Comments to the Author

Reviewer #1: The idea and objective is important. The sample reached is significantly important. However, the panel analysis of the companies lacks proper formation and interpretation falls short of expectations. The general economic view of internationalisation and improtance of economic gain for exports in national currency was used withour proper scientific support. Tehrefore results and suggestions are also too general.

To emphasize again, the onjective and processing is important especially for china but the analytical process and inferences need to be overviewed. The exchange rates of significant trade partners' or at lease usd itself could be considered as an explanatory.

Reviewer #2: Dear Author,

I have listed some constructive comments to criticize the weaknesses of the paper as below.

Constructive Comments:

1. At first glance, the aim, objectives, and methodology of the paper seem a bit vague. There are little adequate clarifications.

2. In introduction section, the authors can reflect the relevant literature review in a more systematic way. In the current form, there is a great mess in the paper. In addition, the follow-up questions can be listed in the introduction to attract the attention of the readers.

3. I recommend to the authors to clarify their methodology in detail, making sure that their planned methods/research tools are fully detailed. They ought to give attention to justifying the chosen methodology in terms of demonstrating applicability, adjustment, and usefulness in the paper.

4. Theory-practice nexus is a good tool for a “thesis vs. counter-thesis = synthesis” approach. In this context, the way how you support and proof your argument and falsify your counterarguments can be more effective to keep up being focused on your story. So that the central research questions and the “argument vs. counter-arguments interactions” are streamlined in a way that addresses the research inquiries in a systematic and effective manner.

5. Referencing needs to be improved very carefully. The authors can enrich the literature review section through adding some up-to-date and relevant scientific materials.

6. I strongly recommend the researchers to detect their scientific work via iThenticate / CrossCheck or Turnitin plagiarism software. Regarding the improvement of the quality of a scientific work in frame of scientific & ethical criteria, my advice to the authors would be to register to some free online interactive e-training courses which are listed as such: i) Making sense of science stories; ii) Publishers: Origins, roles and contributions; iii) The journal publishing cycle; iv) How do Editors look at your paper?; v) Preparing your manuscript; vi) Structuring your article; vii) Successful grant applications getting it right; viii) How to review a manuscript; ix) How to get published structuring your article; and x) Ethics responsibilities for Reviewers. The authors will learn about much more information (e.g., salami slicing, least publishable units, plagiarism, falsification, duplication, fabrication and so on) which are associated with preparing an excellent manuscript for a peer review journal.

7. In fact, I appreciate there has been a lot of reading and ground covered, but this will not appeal to readers if the paper lacks a strong focus, compelling argument and discussion, and an indication of why the paper holds value to the readership of PLOS ONE.

Some Technical Critiques and Recommendations for the Author:

1) The author should strictly adhere to submission guidelines of the journal.

2) There are several grammatical errors and instances of badly worded/constructed sentences. Please check the manuscript and refine the language carefully. I suggest using proof reading papers before submission.

Reviewer #3: This paper attempts to address the issue concerning China’s efforts to internationalise its currency, Yuan. Good exports volume and export competitiveness promote a country’s own currency. In this case, China’s trade with the world constitutes a larger share of world exports, of which China is using its RMB to settle cross-border trade agreements. This also signifies the growing importance of China’s Renminbi (RMB) as an important currency of global trade and commerce.

The paper is analytic and informative, and stresses on the export competitiveness and the impact it has on internalisation of RMB. The authors have engaged in an analytical research and have used extensive datasets to test their hypothesis. It also stresses the role of currency in trade settlement across international borders. The authors have duly reviewed the literature and have drawn examples citing past studies on this topic.

The statistical analyses thus performed are robust, with multimodal framework designed to test the role of export competitiveness on promoting a nation’s currency. The theoretical analyses address both the microeconomic level and macro level picture depicting the choice of currency in cross-border trade settlement.

The authors have provided the datasets based upon which they have performed rigorous statistical analyses. They have also performed robustness test in line with the main objective of the paper to validate the hypotheses.

The authors firmly stuck to the core analysis to figure out what factors could strengthen cross-border used of RMB as means for settling cross- border trades and transactions. It has shed more knowledge on this frontier which could be taken up for further analysis to validate the theory of export competitiveness and how it promotes a country’s own currency, in this case, China.

6. PLOS authors have the option to publish the peer review history of their article (what does this mean? ). If published, this will include your full peer review and any attached files.

**Do you want your identity to be public for this peer review?** For information about this choice, including consent withdrawal, please see our Privacy Policy .

Reviewer #1: No

Reviewer #2: **Yes: ** Dorian Aliu

Reviewer #3: **Yes: ** SIDHARTA CHATTERJEE

---

## [Author Response · Author response to Decision Letter 1]

17 Dec 2024

We have carefully addressed all the comments and suggestions made by the reviewers, which are detailed in the attached "Response to Reviewers" document.

---

## [Decision Letter · Decision Letter 1]

10 Jan 2025

The role of export competitiveness in driving Renminbi cross-border settlement: Micro-level evidence from China

PONE-D-24-43217R1

Dear Dr. JIANG,

We’re pleased to inform you that your manuscript has been judged scientifically suitable for publication and will be formally accepted for publication once it meets all outstanding technical requirements.

Kind regards,

Xiaoyong Zhou, Ph.D.

Academic Editor

PLOS ONE

Additional Editor Comments (optional):

Reviewers' comments:

Reviewer's Responses to Questions

**Comments to the Author**

1. If the authors have adequately addressed your comments raised in a previous round of review and you feel that this manuscript is now acceptable for publication, you may indicate that here to bypass the “Comments to the Author” section, enter your conflict of interest statement in the “Confidential to Editor” section, and submit your "Accept" recommendation.

Reviewer #1: All comments have been addressed

Reviewer #2: All comments have been addressed

Reviewer #3: All comments have been addressed

2. Is the manuscript technically sound, and do the data support the conclusions?

Reviewer #1: Yes

Reviewer #2: Yes

Reviewer #3: Partly

3. Has the statistical analysis been performed appropriately and rigorously? 

Reviewer #1: Yes

Reviewer #2: N/A

Reviewer #3: Yes

4. Have the authors made all data underlying the findings in their manuscript fully available?

Reviewer #1: Yes

Reviewer #2: Yes

Reviewer #3: (No Response)

5. Is the manuscript presented in an intelligible fashion and written in standard English?

Reviewer #1: Yes

Reviewer #2: Yes

Reviewer #3: Yes

6. Review Comments to the Author

Reviewer #1: The paper seems to be throughly revised. Available literature seems to be enriched as well. The lacking information on analyses and inferences were provided. The inference on low r2 may be re-checked of its level can be justified with reference to available literature.

Reviewer #2: Dear Author,

The paper seems like an interesting job. Thus, I consider the paper suitable for being published in the PLOS ONE. I support this research that is deemed a good contribution to the journal’s future achievements. The revision that has been made by the authors is satisfactory. My decision is to accept the paper. Apparently, the current version of the manuscript has convinced me. I affirm the revised version. To me, no further revision is needed.

I wish to express my sincere appreciation to the authors for their scientific efforts during the revision process. Great job. Congratulations!

Reviewer #3: The authors have diligently addressed the issues raised in the previous review. Th paper seems appropriate for publication, as it will stimulate further research in this domain, and since not many papers have addressed similar topic related to export competitiveness and its impact on exchange rate and domestic currency, in this case, Chinese Renminbi. Similar research may be conducted by other countries to asses the nature and levels of their individual export competitiveness and strength of their own currency that are used for cross-border settlement.

7. PLOS authors have the option to publish the peer review history of their article (what does this mean? ). If published, this will include your full peer review and any attached files.

**Do you want your identity to be public for this peer review?** For information about this choice, including consent withdrawal, please see our Privacy Policy .

Reviewer #1: No

Reviewer #2: **Yes: ** Dorian Aliu

Reviewer #3: **Yes: ** SIDHARTA CHATTERJEE

---

## [Editor Report · Acceptance letter]

PONE-D-24-43217R1

PLOS ONE

Dear Dr. JIANG,

I'm pleased to inform you that your manuscript has been deemed suitable for publication in PLOS ONE. Congratulations! Your manuscript is now being handed over to our production team.

Kind regards,

on behalf of

Dr. Prof. Xiaoyong Zhou

Academic Editor

PLOS ONE